# Association of the interatrial block and left atrial fibrosis in the patients without history of atrial fibrillation

**Arintaya Phrommintikul[1,2], Natnicha Pongbangli[3], Siriluck Gunaparn[1], Narawudt Prasertwitayakij[1], Teerapat Nantsupawat[1], Wanwarang Wongcharoen[1]\***

**1** Division of Cardiology, Department of Internal medicine, Chiang Mai University, Chiang Mai, Thailand,
**2** Center for Medical Excellence, Faculty of Medicine, Chiang Mai University, Chiang Mai, Thailand,
**3** Division of Cardiology, Department of Internal Medicine, Chiangrai Prachanukroh Hospital, Chiang Rai, Thailand

\* bwanwarang@yahoo.com

**Data Availability Statement:** Association of the interatrial block and LA fibrosis in the patients without history of AF does not cover data posting

## Abstract

Presence of left atrial (LA) fibrosis reflects underlying atrial cardiomyopathy. Interatrial block (IAB) is associated with LA fibrosis in patients with atrial fibrillation (AF). The association of IAB and LA fibrosis in the patients without history of AF is unknown. We examined association of IAB and LA fibrosis in the patients without AF history. This is a retrospective analysis of 229 patients undergoing cardiac magnetic resonance imaging (CMR). LA fibrosis was reported from spatial extent of late gadolinium enhancement of CMR. IAB was measured from 12-lead electrocardiography using digital caliper. Of 229 patients undergoing CMR, prevalence of IAB was 50.2%. Patients with IAB were older (56.9±13.9 years vs. 45.9±19.2 years, p<0.001) and had higher prevalence of co-morbidities. Left ventricular ejection fraction was lower in IAB group. LA volume index (LAVI) was greater in IAB group (54.6±24.9 ml/m² vs. 43.0±21.1 ml/m², p<0.001). Patients with IAB had higher prevalence of LA fibrosis than those without IAB (70.4% vs. 21.2%; p<0.001). After multivariable analysis, only IAB and LAVI were independent factors that predict LA fibrosis. Prevalence of IAB in patients undergoing CMR was high. IAB was highly associated with LA fibrosis and larger LA size in patients without AF history.

## Introduction

Atrial cardiomyopathy is defined by a complex of structural, architectural, contractile or electrophysiological changes affecting the atrium with the potential to produce clinically relevant manifestations [1]. It is related to an increased risk of thromboembolism independent of the presence of atrial fibrillation (AF) [2]. Left atrial (LA) fibrosis is the hallmark of LA structural remodeling [3] and serves as a substrate for slow conduction, intra-atrial re-entry, predisposing to future atrial arrhythmia [4, 5].

A prior study has shown that LA fibrosis is correlated with an elevated risk of stroke, heart failure [3, 6] and diastolic dysfunction in patients without AF [7]. In addition, it predicts the onset of new AF and the recurrence of after ablation [7].

in public databases. However, data are available upon request should be sent to the ethics committee of Faculty of Medicine, Chiang Mai University (researchmed@cmu.ac.th).

**Funding:** The author(s) received no specific funding for this work.

**Competing interests:** The authors have declared that no competing interests exist.

Currently, cardiac magnetic resonance with late gadolinium enhancement (LGE-CMR) stands as the gold standard for imaging fibrosis [8]. Late enhancement CMR, using gadolinium contrast, has been shown to localize and measure the extent of structural remodeling or fibrosis linked to AF in the LA [9–11]. Additionally, LGE-CMR has demonstrated its utility in locating and measuring scar formation in the LA after radiofrequency ablation [10–13]. Despite these advancements, the routine utilization of LA imaging for diagnosis and risk stratification remains limited [7].

Interatrial block (IAB) is defined as prolonged conduction time between the right atrium and the LA, leading to impulse delay or blockage and resulting in a prolonged P-wave duration ($\geq$120 milliseconds) observed on a 12-lead electrocardiogram (ECG) [14]. Similar to other conduction delays, IAB can be classified into partial and advanced IAB.(14) The classification is based on the P-wave duration and more significantly, the P-wave morphology in the inferior leads [15]. P-wave duration $\geq$120 milliseconds with positive P wave morphology indicates the partial IAB, signifying delayed inter-atrial conduction via Bachmann's bundle [14]. P-wave duration $\geq$120 milliseconds with biphasic (positive-negative) P-wave morphology designates the advanced IAB, representing complete degree of interatrial blockade at Bachmann's bundle and resulting in caudocranial activation in the atria [14].

The connection between LA remodeling and P wave abnormalities has been documented in patients with history of AF [16]. Previous study has indicated that P wave dispersion, defined as the difference between the widest and the narrowest P-wave duration, is not a reliable predictor of the presence of LA fibrosis [16]. On the other hand, abnormal P-wave terminal force in lead V1 (PTFV1) has been identified as an independent predictor for atrial electrical dysfunction, though not for structural remodeling [17]. A recent study suggests that advance IAB is notably linked to the presence of LA fibrosis in patient undergoing AF ablation [18].

However, in the patients without history of AF, the association between IAB and atrial fibrosis had not been explored. Therefore, our study aimed to evaluate the association between IAB and LA fibrosis using late gadolinium enhancement imaging via CMR.

## Methods

### The studied population

This is a retrospective analysis of patients who presented at Maharaj Nakhon Chiang Mai hospital for CMR during the specified study periods. The inclusion criteria comprised individuals who were (1) aged over 18 years; (2) undergoing CMR for various indications between April 1, 2013 to December 1, 2021; (3) had a standard 12-lead electrocardiogram (ECG) conducted within 6 months before or after of the CMR. Patients with history of AF, sinus arrest or those in whom P wave could not be identified from the standard 12-lead ECG were excluded. Demographic characteristics were collected. The study protocol was approved by the ethics committee of Faculty of Medicine, Chiang Mai University on 26 January 2022. The data were evaluated for research purposes from February 2023 to June 2023 (Fig 1).

**Ethics approval and consent to participate.** Association of the interatrial block and LA fibrosis in the patients without history of AF was approved by the ethics committee of the Faculty of Medicine, Chiang Mai University, approval number 032/2565 and was registered in thaiclinicaltrials.org, identification number TCTR20231003004. The investigations were carried out following the Declaration of Helsinki, including written informed consent from all participants.

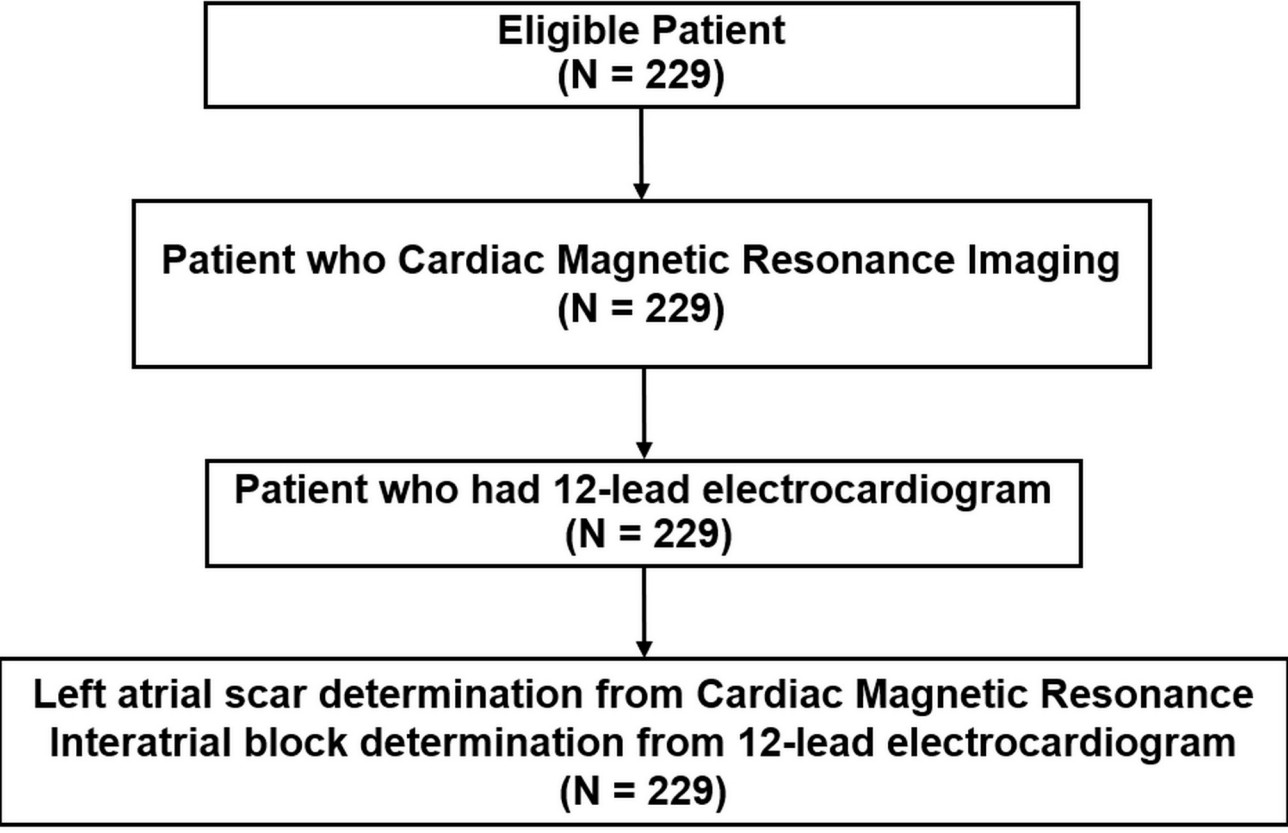

**Fig 1. Consort flow diagram.**

### ECG analysis

The 12-lead ECG criteria for partial IAB were established as a P-wave duration of ≥120 milliseconds with a positive P wave morphology in the inferior leads. For advanced IAB, ECG criteria were defined as a P wave of ≥120 milliseconds with a biphasic (positive-negative) P-wave morphology in the inferior leads.

P wave duration and morphology were manually analyzed from the standard 12-lead ECG using a 150 Hz filter, a paper speed of 25 mm/s, and an amplitude of 10 mm/mV. P-wave duration was defined as the interval between the onset and offset of P wave in the inferior leads. Two independent observers conducted P-wave measurements. In cases where there was a discrepancy between the two independent measurements, a third independent observer made the final determination.

### CMR analysis

Data from CMR imaging conducted between April 1, 2013, and December 1, 2021, were collected from the electronic medical record. CMR was performed using clinical 1.5 T CMR scanner (General Electric CV/i, Milwaukee, USA) and a 6-channel phased array body coil in combination with a 6-channel spine matrix coil. An ECG-gated, breath-holding cine CMR images were acquired in the short axis along left ventricle from tip of left ventricular apex to mitral valve annulus, long axis two-, three- and four-chamber views by Steady-State Precession sequence with the following parameters: TE/TR 3.0/1.5 ms; flip angle 78˚; in-plane pixel size

1.5×1.5 mm2; slice thickness 8 mm; 30 frames per ECG R-R interval. LGE images were acquired within 10–15 minutes following the injection of gadopentetate dimeglumine (0.2 mmol/kg, Bayer Healthcare Pharmaceuticals, Montville, NJ) using short axis imaging Inversion-recovery gradient echo technique (TR 7.1 ms; TE 3.1 ms; TI individually determined to null the myocardial signal, range 180–250 ms, slice thickness 15 mm, matrix 256 × 192, number of acquisitions = 2).

LA volume was assessed from 2-chamber view and 4-chamber view from Steady-State Precession sequence. Maximal LA volume was LA volume at end-systole, immediately before mitral valve opening. LA fibrosis was manually estimated using 2-chamber, 3-chamber and 4-chamber views from LGE image during atrial end-diastole.

The LA wall area was delineated through manual tracing. A threshold criterion for identifying the presence of LGE was applied, considering signal intensity exceeding two standard deviations above that of a normative myocardium within the corresponding image. The quantification of LA fibrosis was established as the percentage of LGE within the LA. The categorization of LA fibrosis severity was divided into mild ($<$35% LGE), moderate (35–69% LGE), and severe ($\geq$70% LGE) based on the extent of involvement within the LA myocardium. The CMR imaging analyses were interpreted by 2 experienced investigators who were blinded to other results.

## Statistical analysis

The categorical data were presented as N (%) and compared between groups with Fisher's exact test. The continuous data were presented as mean ± SD or median (interquartile range) and compared between groups with Student's t-test or Mann-Whitney U test where appropriate. The univariable risk factors with p-value < 0.1 were included into multivariable analysis. P value < 0.05 was considered statistical significance. Statistical software package IBM SPSS Statistics for Windows, version 23.0 (IBM Corp., Armonk, NY, USA, https://www.ibm.com/products/spss-statistics) was used for analysis.

## Results

A total of 229 patients who underwent CMR imaging were included in the study. The most common reason for CMR scanning was to assess myocardial ischemia in patients with known or suspected coronary artery disease (45%). The other indications included known or suspected cardiomyopathy (26%), suspected myocarditis (12%), congenital heart disease (11%), ventricular arrhythmia/ premature ventricular complex (5%) and cardiac mass (1%).

The prevalence of IAB was 50.2% (115 out of 229 patients) comprised of partial IAB (46%) and advanced IAB (4.2%). The clinical characteristics of patients according to the presence of IAB are reported in Table 1. Patients with IAB were older than patients without IAB (56.9 ± 13.9 years vs. 45.9±19.2 years, p<0.001). Left ventricular ejection function was lower in patients with IAB compared to patients without IAB (45.1±18.6% vs. 53.8±17.9%, p <0.001). LA volume index (LAVI) was greater in IAB group (54.6±24.9 ml/m$^2$ vs. 43.0±21.1 ml/m$^2$, p<0.001). The clinical characteristics were not different between patients with partial IAB and those with advanced IAB.

LA fibrosis was presented in 135 (59%) of 229 patients. Among those with LA fibrosis, the mean percentage of atrial fibrosis extent was 68.6±25.4%. Twenty-three (10.0%) patients had a mild degree of atrial fibrosis with a mean extent of 21.5±4.3%, 49 (21.4%) patients had a moderate degree of atrial fibrosis with a mean extent of 44.6±7.9% and 63 (27.5%) patients had a severe degree of atrial fibrosis with a mean extent of 84.2±13.6%. The clinical characteristics between the patients with and without LA fibrosis from CMR are summarized in Table 2.

**Table 1. Clinical characteristics of patients with patients with IAB and without IAB.**

| Baseline characteristics | No IAB (49.7%) (N = 114) | Total IAB (51.3%) (N = 115) | Partial IAB (45.8%) (N = 105) | Advanced IAB (5.5%) (N = 10) | P value* |
|---|---|---|---|---|---|
| Age (years) | 45.9±19.2 | 56.9 ± 13.9 | 56.6 ± 13.8 | 60.8 ± 15.6 | <0.001 |
| BMI (kg/m2) | 22.9±4.3 | 22.4±4.5 | 22.2±4.4 | 23.9±5.1 | 0.394 |
| Male | 67 (58.8%) | 77 (67.0%) | 71 (67.6%) | 6 (60.0%) | 0.220 |
| Diabetic mellitus | 18 (15.8%) | 37 (32.2%) | 33 (31.4%) | 4 (40.0%) | 0.005 |
| Hypertension | 35 (30.7%) | 51 (44.3%) | 46 (43.8%) | 5 (50.0%) | 0.041 |
| Dyslipidemia | 31 (27.2%) | 39 (33.9%) | 36 (34.3%) | 3 (30.0%) | 0.316 |
| Stroke | 5 (4.4%) | 5 (4.3%) | 4 (3.8%) | 1 (10.0%) | 1.000 |
| Coronary artery disease | 29 (25.4%) | 58 (50.4%) | 52 (49.5%) | 6 (60%) | <0.001 |
| Heart failure | 23 (20.2%) | 36 (31.3%) | 34 (32.4%) | 2 (20.0%) | 0.069 |
| Chronic kidney disease | 10 (8.8%) | 16 (13.9%) | 13 (12.4%) | 3 (30.0%) | 0.298 |
| Valvular heart disease | 32 (28.1%) | 41(35.7%) | 39 (37.1%) | 2 (20.0%) | 0.257 |
| Dilated cardiomyopathy | 16 (14%) | 21 (18.3%) | 18 (17.1%) | 3 (30.0%) | 0.473 |
| ACEI/ARB | 35 (30.7%) | 61 (53%) | 56 (53.3%) | 5 (50.0%) | 0.001 |
| Beta-blocker | 62 (54.4%) | 71 (61.7%) | 63 (60.0%) | 8 (80.0%) | 0.285 |
| Spironolactone | 24 (21.1%) | 32 (27.8%) | 30 (28.6%) | 2 (20.0%) | 0.282 |
| LVEF (%) | 53.8 ± 17.9 | 45.1± 18.6 | 45.1±19.1 | 45.5±13.9 | <0.001 |
| LAVI (ml/m2) | 43.0 ± 21.1 | 54.6 ± 24.9 | 54.2±25.2 | 58.8±23.4 | <0.001 |

* Compared between No IAB and total IAB Data are presented as mean ± SD or N (%). ACEI: Angiotensin-converting enzyme inhibitor, ARB: Angiotensin receptor blocker; BMI: Body mass index, LAVI: left atrial volume index, LVEF: left ventricular ejection function

Patients with LA fibrosis were older than patients without LA fibrosis (54.5±15.6 years vs 47.1 ±19.5 years, p = 0.002). The prevalence of diabetes mellitus, hypertension and coronary artery disease was higher in patients with LA fibrosis (29.6% vs. 16%; p = 0.019, 44.4% vs 27.7%; p = 0.012 and 47.4% vs. 24.5%; p = 0.001, respectively.

The left ventricular ejection fraction was significantly lower in patients with LA fibrosis than in patients without (44.9±18.8% vs. 55.9±16.7%, p < 0.001). The LA volume index was significant higher in patients with LA fibrosis than in those without LA fibrosis (57.0±24.3mL/m$^2$ vs. 33.6±13.5mL/m$^2$, p < 0.001). Importantly, the presence of IAB was more prevalent in patients with LA fibrosis than in those without LA fibrosis (70.4% vs. 21.2%; p<0.001). According to the varying extent of LA fibrosis, IAB was observed in 56.5%, 71.4%, and 74.7% of patients with mild, moderate, and severe LA fibrosis, respectively.

Factors associated with left atrial fibrosis from univariable and multivariable analyses are presented in Table 3. The multivariable analysis revealed that a larger LA volume index and the presence of IAB were the two independent predictors of LA fibrosis assessed by CMR.

## Discussion

Atrial fibrosis and myofibril disorganization constitute the characteristic phenotype of atrial cardiomyopathy [1]. Previous studies have demonstrated the association of atrial fibrosis with an increased risk of stroke, heart failure, and the prediction of new-onset AF and AF recurrence after ablation [3, 6, 7]. CMR-LGE is considered the gold standard in imaging fibrosis [8]. However, the widespread adoption of CMR is limited due to its cost, requirement for higher technical expertise, and relatively long time for image acquisition [7]. In contrast, 12-lead ECG is inexpensive test, more widely available and does not require technical expertise. The utility of 12-lead ECG in predicting LA fibrosis is valuable in clinical practice for identifying patients at risk.

**Table 2. Clinical characteristics of patients with and without left atrial fibrosis.**

| Baseline characteristics | No LA fibrosis (41%) (N = 94) | All LA fibrosis (59%) (N = 135) | Mild LA fibrosis (10%) (N = 23) | Moderate LA fibrosis (21.4%) (N = 494) | Severe LA fibrosis (27.5%) (N = 63) | P value* |
|---|---|---|---|---|---|---|
| Age (years) | 47.1±19.5 | 54.5±15.6 | 51.9±16.0 | 54.9±15.9 | 55.0±15.4 | 0.002 |
| BMI (kg/m2) | 23.1±4.4 | 22.3±4.4 | 23.1±4.4 | 22.9±3.7 | 21.5±4.9 | 0.164 |
| Male | 56(59.6%) | 88(65.2%) | 9(75.0%) | 23(67.6%) | 56(62.9%) | 0.407 |
| Diabetic mellitus | 15(16%) | 40(29.6%) | 8(34.8%) | 13(26.5) | 19(30.2%) | 0.019 |
| Hypertension | 26(27.7%) | 60(44.4%) | 6(26.1%) | 24(49.0%) | 30(47.6%) | 0.012 |
| Dyslipidemia | 28(29.8%) | 42(31.1%) | 7(30.4%) | 13(26.5%) | 22(34.9%) | 0.885 |
| Stroke | 4(4.3%) | 6(4.4%) | 0(0%) | 1(2%) | 5(7,.9%) | 1.000 |
| Coronary artery disease | 23(24.5%) | 64(47.4%) | 9(39.1%) | 25(51.0%) | 30(47.6%) | 0.001 |
| Heart failure | 17(18.1%) | 42(31.1%) | 6(26.1%) | 17(34.7%) | 19(30.2%) | 0.032 |
| Chronic kidney disease | 9(9.6%) | 17(12.6%) | 2(8.7%) | 7(14.3%) | 8(12.7%) | 0.531 |
| Valvular heart disease | 18(19,1%) | 55(40.7%) | 7(30.4%) | 17(34.7%) | 31(49.2%) | < 0.001 |
| Dilated cardiomyopathy | 16(17%) | 21(15.6%) | 2(8.7%) | 6(12.2%) | 13(20.6%) | 0.856 |
| ACEI/ARB | 33(35.1%) | 63(46.7%) | 7(30.4%) | 24949.0%) | 32(50.8%) | 0.102 |
| Beta-blocker | 51(54.3%) | 82(60.7%) | 15(65.2%) | 28(57.1%) | 39(61.9%) | 0.343 |
| Spironolactone | 14(14.9%) | 42(31.1%) | 6(26.1%) | 12(24.5%) | 24(38.1%) | 0.006 |
| LVEF (%) | 55.9±16.7 | 44.9±18.8 | 48.3±19.8 | 43.7±18.4 | 44.5±18.9 | <0.001 |
| LAVI (ml/m2) | 33.6±13.5 | 57.0±24.3 | 49.8±185 | 51.1±20.0 | 63.7±27.4 | <0.001 |
| Presence of IAB | 20(21.2%) | 95(70.4%) | 13(56.5%) | 35(71.4%) | 4774.6%) | <0.001 |

Compared between no LA fibrosis and all LA fibrosis. Data are presented as mean ± SD or N (%) ACEI: Angiotensin-converting enzyme inhibitor, ARB: Angiotensin receptor blocker, BMI: Body mass index, LAVI: left atrial volume index; IAB: interatrial block, LVEF: left ventricular ejection function

The prevalence of LA fibrosis in our studied population was 59%, comparable to the reported prevalence of 54% in individuals aged greater than 75 years, with or without AF [19]. The high prevalence of atrial fibrosis observed in our study may be attributed to the significant cardiac conditions present in the majority of recruited patients, including coronary artery disease and cardiomyopathies. The presence of LA remodeling is well-established in patients with cardiovascular diseases. In addition, aging and cardiovascular risk factors are associated with

**Table 3. Univariable and multivariable analyses of factors associated with left atrial fibrosis.**

| | Univariate | | Multivariate | |
|---|---|---|---|---|
| | Risk ratio (95%CI) | P value | Risk ratio (95%CI) | P value |
| Age | 1.02(1.01–1.04) | 0.002 | 0.99(0.95–1.02) | 0.358 |
| Diabetic mellitus | 2.22(1.14–4.31) | 0.019 | 0.73(0.20–2.66) | 0.634 |
| Hypertension | 2.09(1.19–3.68) | 0.012 | 2.35(0.80–6.90) | 0.119 |
| Coronary artery disease | 2.78(1.56–4.97) | 0.001 | 1.05(0.31–3.54) | 0.932 |
| Spironolactone | 2.58(1.31–5.07) | 0.006 | 0.81(0.21–3.16) | 0.759 |
| LVEF | 0.97(0.95–0.98) | <0.001 | 0.98(0.95–1.01) | 0.211 |
| LA volume index (ml/m2) | 1.08(1.05–1.11) | <0.001 | 1.07(1.03–1.11) | <0.001 |
| IAB | 8.79(4.74–16.28) | <0.001 | 5.98(2.43–12.70) | <0.001 |

LVEF: left ventricular ejection function; LAVI: left atrial volume index; IAB: interatrial block

increased inflammation, endothelial dysfunction, and impaired cardiomyocyte function which account for atrial remodeling [2].

The association between left atrial volume and the extent of left atrial fibrosis has been previously reported in patients undergoing AF ablation [20, 21]. Similarly, our study found that the patients with LA fibrosis had a larger LA volume index than those without LA fibrosis. LA dilatation reflects increased wall tension due to elevated LA pressure, serving as a marker for the severity and chronicity of left ventricular dysfunction [22].

A previous study has demonstrated that the presence of IAB is associated with larger LA volumes and atrial fibrosis in the patients with paroxysmal AF [18, 23]. Nevertheless, the association between IAB and atrial fibrosis in patients without AF has been scarcely investigated. To the best of our knowledge, we have demonstrated for the first time the strong association between IAB and moderate to severe LA fibrosis detected by CMR in patients without history of AF.

Importantly, our results demonstrated that IAB was found in half of the patients, with high prevalence of coronary artery disease and cardiomyopathies, despite the absence of a history of AF. The strong association of IAB with LA fibrosis shown in our study underscores the importance of 12-lead ECG screening in the predominantly high cardiovascular risk population. Detecting IAB can help identify the high-risk subset of patients with atrial cardiomyopathy. Emphasizing aggressive therapy targeting modifiable risk factors should be considered to potentially improve atrial remodeling and lead to better cardiovascular outcomes [24–26].

## Limitations

Currently, there is no consensus for the assessment of atrial fibrosis by CMR analysis, potentially leading to variations in prevalence and significance of LA fibrosis across studies. This study constitutes a single-center analysis of patients referred for CMR, introducing a non-negligible chance of selection bias. Larger prospective studies are warranted to confirm our results.

## Conclusions

The prevalence of IAB in patients without a history of AF who had undergone CMR was relatively high. We demonstrated a strong association between the presence of IAB and LA fibrosis in patients without a history of AF.

## Supporting information

**S1 File. Protocol left atrial abnormality V1.0 date 27SEP2021.**
(PDF)

## Author Contributions

**Conceptualization:** Arintaya Phrommintikul, Wanwarang Wongcharoen.

**Data curation:** Arintaya Phrommintikul, Natnicha Pongbangli, Siriluck Gunaparn, Narawudt Prasertwitayakij, Teerapat Nantsupawat.

**Formal analysis:** Arintaya Phrommintikul, Wanwarang Wongcharoen.

**Investigation:** Natnicha Pongbangli, Narawudt Prasertwitayakij, Teerapat Nantsupawat.

**Methodology:** Natnicha Pongbangli.

**Project administration:** Siriluck Gunaparn.

**Supervision:** Narawudt Prasertwitayakij, Teerapat Nantsupawat.

**Validation:** Wanwarang Wongcharoen.

**Writing – original draft:** Arintaya Phrommintikul.

**Writing – review & editing:** Wanwarang Wongcharoen.

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
