## [Decision Letter · Decision Letter 0]

26 Dec 2023

PONE-D-23-31438Association of the interatrial block and left atrial fibrosis in the patients without history of atrial fibrillationPLOS ONE

Dear Dr. Wongcharoen,

Thank you for submitting your manuscript to PLOS ONE. After careful consideration, we feel that it has merit but does not fully meet PLOS ONE’s publication criteria as it currently stands. Therefore, we invite you to submit a revised version of the manuscript that addresses the points raised during the review process.

Dear authors,

Please, address all comments made by the reviewer.  Additionally, include data on LA fibrosis mass and clinical implications of your findings.

We look forward to receiving your revised manuscript.

Kind regards,

Roberto Magalhães Saraiva, MD, PhD

Academic Editor

PLOS ONE

https://onlinelibrary.wiley.com/doi/10.1111/j.1540-8167.2010.01876.x

In your revision ensure you cite all your sources (including your own works), and quote or rephrase any duplicated text outside the methods section. Further consideration is dependent on these concerns being addressed.

4. In the online submission form you indicate that your data is not available for proprietary reasons and have provided a contact point for accessing this data. Please note that your current contact point is a co-author on this manuscript. According to our Data Policy, the contact point must not be an author on the manuscript and must be an institutional contact, ideally not an individual. Please revise your data statement to a non-author institutional point of contact, such as a data access or ethics committee, and send this to us via return email. Please also include contact information for the third party organization, and please include the full citation of where the data can be found.

Additional Editor Comments:

Dear authors,

Please, address all comments made by the reviewer. Additionally, include data on LA fibrosis mass and clinical implications of your findings.

Reviewers' comments:

Reviewer's Responses to Questions

**Comments to the Author**

1. Is the manuscript technically sound, and do the data support the conclusions?

Reviewer #1: Yes

2. Has the statistical analysis been performed appropriately and rigorously? 

Reviewer #1: Yes

3. Have the authors made all data underlying the findings in their manuscript fully available?

Reviewer #1: Yes

4. Is the manuscript presented in an intelligible fashion and written in standard English?

Reviewer #1: No

5. Review Comments to the Author

Reviewer #1: The authors study the association of interatrial block and left atrial fibrosis in a retrospective analysis of 229 patients without history of atrial fibrillation.

Please describe in detail how LA fibrosis evaluated and why a dichotomous variable (fibrosis/no) fibrosis was used as in methods the authors state that the extent of LA fibrosis was divided in to mild, moderate and severe.

Figure 1 is unclear, please include N in all the entries.

Please give data regarding partial and advanced IAB.

English revision is needed.

6. PLOS authors have the option to publish the peer review history of their article (what does this mean?). If published, this will include your full peer review and any attached files.

Reviewer #1: **Yes: **Manuel Martinez-Selles

---

## [Author Response · Author response to Decision Letter 0]

5 Jan 2024

Response: We have checked and ensured that the manuscript meets PLOS ONE’s style requirements.

https://onlinelibrary.wiley.com/doi/10.1111/j.1540-8167.2010.01876.x

In your revision ensure you cite all your sources (including your own works), and quote or rephrase any duplicated text outside the methods section. Further consideration is dependent on these concerns being addressed.

Response: We have revised our manuscript to avoid the overlapping text with previous publication. We have rephrased the duplicated text and have cited one additional reference in the revised manuscript, (Akoum n, et al. JCE 2011 https://onlinelibrary.wiley.com/doi/10.1111/j.1540-8167.2010.01876.x), according to your suggestion. (Page 3, line 57-58) 

Response: Thank you very much for this helpful information. However, our data is not publicly available due to proprietary reasons. We have provided a contact point for accessing our data, as per your suggestion.

4. In the online submission form you indicate that your data is not available for proprietary reasons and have provided a contact point for accessing this data. Please note that your current contact point is a co-author on this manuscript. According to our Data Policy, the contact point must not be an author on the manuscript and must be an institutional contact, ideally not an individual. Please revise your data statement to a non-author institutional point of contact, such as a data access or ethics committee, and send this to us via return email. Please also include contact information for the third-party organization, and please include the full citation of where the data can be found.

Response: Thank you for your suggestion. We have revised the data availability statements in the revised manuscript. The data are available upon request should be sent to the ethics committee of Faculty of Medicine, Chiang Mai University (researchmed@cmu.ac.th). (Page 14, line 238-241)

Response: We have moved the ethics statement to the Methods section, according to your suggestion. (Page 5, line 91-95)

Additional Editor Comments:

Please, address all comments made by the reviewer. Additionally, include data on LA fibrosis mass and clinical implications of your findings.

Response: We have addressed all comments made by the reviewer. We have also incorporated detailed data on LA fibrosis in the Result section page 9, line 161-169 and Table 2 of the revised manuscript. (Page 10)

The clinical implications of our findings have been demonstrated in Discussion section as follows; “The strong association of IAB with LA fibrosis shown in our study underscores the importance of 12-lead ECG screening in the predominantly high cardiovascular risk population. Detecting IAB can help identify the high-risk subset of the patients with atrial cardiomyopathy. Emphasizing aggressive therapy targeting modifiable risk factors should be considered to potentially improve atrial remodeling and lead to better cardiovascular outcomes.” (Page 13, line 223-227)

Reviewer #1: The authors study the association of interatrial block and left atrial fibrosis in a retrospective analysis of 229 patients without history of atrial fibrillation.

Please describe in detail how LA fibrosis evaluated and why a dichotomous variable (fibrosis/no) fibrosis was used as in methods the authors state that the extent of LA fibrosis was divided in to mild, moderate and severe.

Response: We appreciate the reviewer’s comments. We evaluated LA fibrosis by manual tracing the left atrial wall. A threshold criterion for identifying the presence of LGE was applied, considering signal intensity exceeding two standard deviations above that of a normative myocardium within the corresponding image. The information was added in the Methods section (Page 6, line 122-124) and it reads “The LA wall area was delineated through manual tracing. A threshold criterion for identifying the presence of LGE was applied, considering signal intensity exceeding two standard deviations above that of a normative myocardium within the corresponding image. The quantification of LA fibrosis was established as the percentage of LGE within the LA. The categorization of LA fibrosis severity was divided into mild (<35% LGE), moderate (35-69% LGE), and severe (≥70% LGE) based on the extent of involvement within the LA myocardium.” 

We utilized a dichotomous variable (no fibrosis/fibrosis) for LA fibrosis due to the absence of a consensus on cut-off criteria for LA fibrosis extent and its clinical significance. Nonetheless, additional data on the three varying extents of LA fibrosis is provided in the Results section page 9, line 161-169 and Table 2 in the revised manuscript. (Page 10)

Figure 1 is unclear, please include N in all the entries.

Response: We apologize for the unclear figure. We have included the number of patients in Figure 1 in the revised manuscript. All the 229 eligible patients in our study underwent cardiac MRI and 12-lead ECG. As a result, the association between LA scar from cardiac MRI and interatrial block from 12-lead ECG was examined in all 229 patients.

Please give data regarding partial and advanced IAB.

Response: We have provided the data regarding partial and advanced IAB in Table 1 in the revised manuscript. (Page 8)

English revision is needed.

Response: The English revision has been incorporated into the revised manuscript by a native English speaker, in accordance with your suggestion.

---

## [Decision Letter · Decision Letter 1]

16 Jan 2024

Association of the interatrial block and left atrial fibrosis in the patients without history of atrial fibrillation

PONE-D-23-31438R1

Dear Dr. Wongcharoen,

We’re pleased to inform you that your manuscript has been judged scientifically suitable for publication and will be formally accepted for publication once it meets all outstanding technical requirements.

Kind regards,

Roberto Magalhães Saraiva, MD, PhD

Academic Editor

PLOS ONE

Additional Editor Comments (optional):

Reviewers' comments:

Reviewer's Responses to Questions

**Comments to the Author**

1. If the authors have adequately addressed your comments raised in a previous round of review and you feel that this manuscript is now acceptable for publication, you may indicate that here to bypass the “Comments to the Author” section, enter your conflict of interest statement in the “Confidential to Editor” section, and submit your "Accept" recommendation.

Reviewer #1: All comments have been addressed

2. Is the manuscript technically sound, and do the data support the conclusions?

Reviewer #1: Yes

3. Has the statistical analysis been performed appropriately and rigorously? 

Reviewer #1: Yes

4. Have the authors made all data underlying the findings in their manuscript fully available?

Reviewer #1: Yes

5. Is the manuscript presented in an intelligible fashion and written in standard English?

Reviewer #1: Yes

6. Review Comments to the Author

Reviewer #1: There is a typo in figure 1. Perhaps a second English revision is a good idea. All my comments have been attended.

7. PLOS authors have the option to publish the peer review history of their article (what does this mean?). If published, this will include your full peer review and any attached files.

Reviewer #1: **Yes: **Manuel Martinez-Selles

---

## [Editor Report · Acceptance letter]

30 Jan 2024

PONE-D-23-31438R1 

PLOS ONE

Dear Dr. Wongcharoen, 

I'm pleased to inform you that your manuscript has been deemed suitable for publication in PLOS ONE. Congratulations! Your manuscript is now being handed over to our production team.

Kind regards, 

on behalf of

Dr. Roberto Magalhães Saraiva 

Academic Editor

PLOS ONE